# A global database of plant services for humankind

**Rafael Molina-Venegas**[1]*, **Miguel Ángel Rodríguez**[1], **Manuel Pardo-de-Santayana**[2,3], **David J. Mabberley**[4,5,6]

1 Universidad de Alcalá, GLOCEE-Global Change Ecology and Evolution Group, Department of Life Sciences, Alcalá de Henares, Spain, 2 Department of Biology (Botany), Universidad Autónoma de Madrid, Madrid, Spain, 3 Centro de Investigación en Biodiversidad y Cambio Global (CIBC-UAM), Madrid, Spain, 4 Wadham College, University of Oxford, Oxford, United Kingdom, 5 Department of Biological Sciences, Macquarie University, Sydney, New South Wales, Australia, 6 Australian Institute of Botanical Science (National Herbarium of New South Wales), Sydney, New South Wales, Australia

* rafmolven@gmail.com

**Data Availability Statement:** The database is available at figshare repository: https://doi.org/10.6084/m9.figshare.13625546.v3.

## Abstract

Humanity faces the challenge of conserving the attributes of biodiversity that may be essential to secure human wellbeing. Among all the organisms that are beneficial to humans, plants stand out as the most important providers of natural resources. Therefore, identifying plant uses is critical to preserve the beneficial potential of biodiversity and to promote basic and applied research on the relationship between plants and humans. However, much of this information is often uncritical, contradictory, of dubious value or simply not readily accessible to the great majority of scientists and policy makers. Here, we compiled a genus-level dataset of plant-use records for all accepted vascular plant taxa (13489 genera) using the information gathered in the 4th Edition of *Mabberley's plant-book*, the most comprehensive global review of plant classification and their uses published to date. From 1974 to 2017 all the information was systematically gathered, evaluated, and synthesized by David Mabberley, who reviewed over 1000 botanical sources including modern Floras, monographs, periodicals, handbooks, and authoritative websites. Plant uses were arranged across 28 standard categories of use following the Economic Botany Data Collection Standard guidelines, which resulted in a binary classification of 9478 plant-use records pertaining human and animal nutrition, materials, fuels, medicine, poisons, social and environmental uses. Of all the taxa included in the dataset, 33% were assigned to at least one category of use, the most common being "ornamental" (26%), "medicine" (16%), "human food" (13%) and "timber" (8%). In addition to a readily available binary matrix for quantitative analyses, we provide a control text matrix that links the former to the description of the uses in *Mabberley's plant-book*. We hope this dataset will serve to establish synergies between scientists and policy makers interested in plant-human interactions and to move towards the complete compilation and classification of the nature's contributions to people upon which the wellbeing of future generations may depend.

**Funding:** This publication was funded by the project "Plant evolutionary history and human well-being in a changing world; assessing theoretical foundations using empirical evidence and new phylogenetic tools" (CM/JIN/2019-005), granted by the Regional Government of Madrid (Consejería de Ciencia, Universidades e Innovación) and Universidad de Alcalá (Spain) to RM-V. RM-V was supported by a TALENTO fellowship of the Regional Government of Madrid (2018-T2/AMB-10332), and MAR by the Spanish Ministry of Science and Innovation (grant CGL2017-86926-P).

**Competing interests:** The authors have declared that no competing interests exist.

## Introduction

Following our failure to fully achieve the 20 Aichi biodiversity targets included in the Strategic Plan for Biodiversity 2011–2020 [1], nations are now working together in developing the post-2020 Global Biodiversity Framework (GBF), an ambitious initiative that will serve as a springboard to prospect the 2050 vision of "living in harmony with nature" [2]. Recognizing all the human benefits that are directly provided by biodiversity is one of the main goals of the post-2020 GBF [3], which aspires to unlock these natural resources and promote basic and applied research on the relationship between humans and the rest of nature [4]. As such, the recent appearance of few cross-disciplinary journals focusing on the interface between biodiversity and society attests a growing interest in the field [5–7].

Among all the organisms that are beneficial to humans, plants stand out as the most important providers of natural services, including not only basic resources but also psychological needs [8], and thus they are often preferred in research projects that lay at the crossroads of biodiversity and human societies [9–11]. However, much of the information on the ethnobotanical and economic use of plants is very sparse and often uncritical, contradictory, of dubious value, or simply not readily accessible to the great majority of scientists and policy makers. Further, transferring available information into a format suitable for quantitative analyses is by no means trivial, and even the most elemental task of distinguishing between types of benefits in the continuum of plant services that are described in the literature requires hard training and dedication. In fact, while ethnobotanists have paid great attention to elaborate disparate classifications of plant benefits [12,13], the use of generalizable classification schemes such as the Economic Botany Data Collection Standard [14] is still rare (but see [9,11]).

Here, we present a global genus-level database of plant-use records that were collated from the 4th Edition of *Mabberley's plant-book* [15]. Although *Mabberley's plant-book* is the most comprehensive global review of plant classification and their uses published to date, the information on plant services it contains has hardly been used in scientific research (but see [11]). This is likely because the book is extraordinarily condensed and plant uses are narratively described rather than sorted into recognizable categories of use. Conscious of the value of this untapped resource, we made an unprecedented effort to manually extract all the information on plant uses from the book and to sort them into standard categories of benefits [14]. We aim at showing the proven value of a treasure-trove of plant-use information that has been curated over more than 40 years and is now readily available for the scientific and policy-making communities.

## Material and methods

The database contains binary information (presence/absence data) on plant-use records for all accepted vascular plant genera described in the 4th Edition of *Mabberley's plant-book* [15], the most comprehensive global review of plant classification and their uses published hitherto. All the information included in *Mabberley's plant-book* was synthesized by David Mabberley from 1974 to 2017, who systematically reviewed over 1000 botanical sources including modern Floras, monographs, periodicals, handbooks, and authoritative websites (all references can be found in [15]). From September 2019 to February 2020, we conducted a double-check manual screening of all plant uses described in *Mabberley's plant-book* and arranged them across 28 standard categories of use following the guidelines in the Economic Botany Data Collection Standard [14] (hereafter "Collection Standard") and our expert criteria. The categories pertained to different dimensions of plant benefits including environmental uses (bioindicators/bioremediators, soil improvers, ornamental, hedging/shelter) human and animal nutrition (human food, human-food additives, vertebrate food, invertebrate food), fuels (fuelwood,

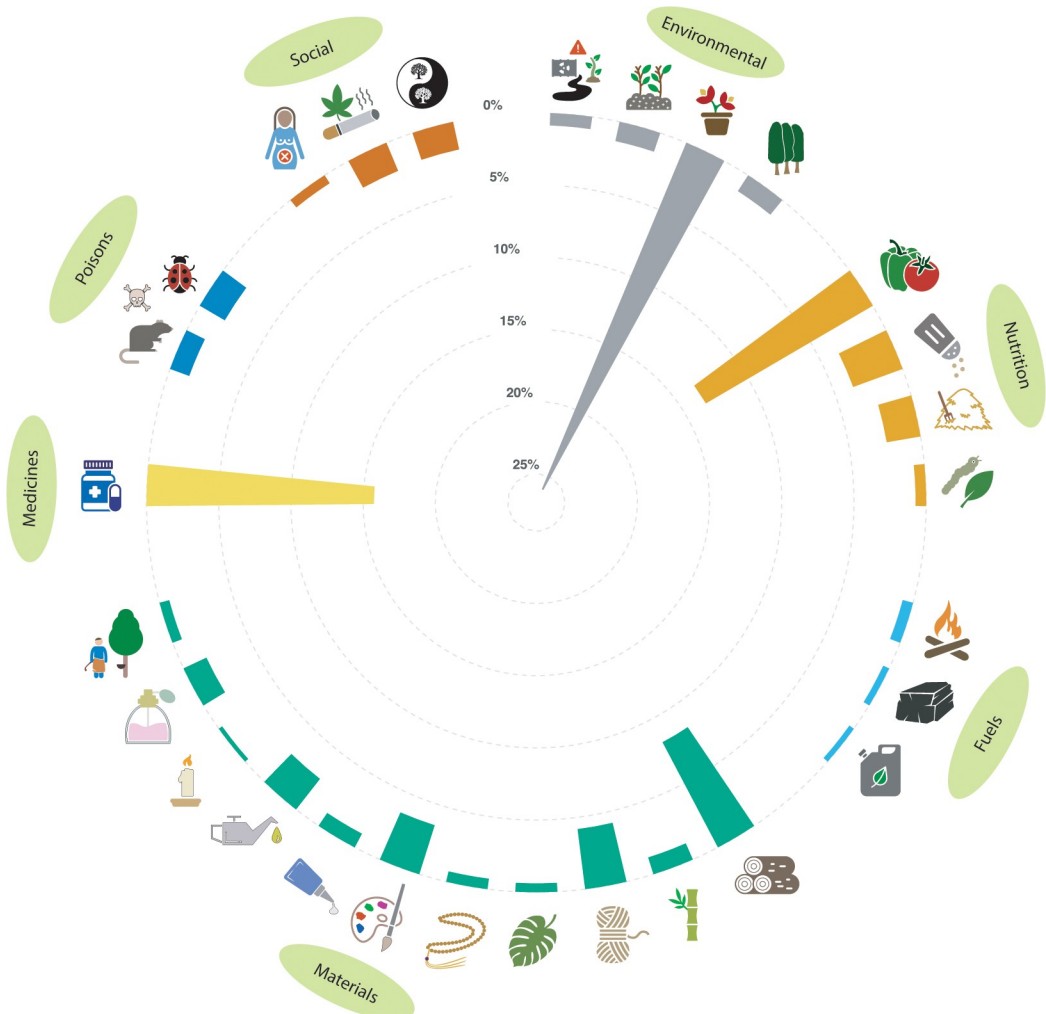

**Fig 1. Proportion of plant-use records per category included in the database.** From twelve o'clock and clockwise: bioindicators and bioremediators (n = 81), soil improvers (n = 129), ornamental (n = 2464), hedges and shelters (n = 123), human food (n = 1269), food additives (n = 345), vertebrate food (n = 225), invertebrate food (n = 69), fuelwood (n = 70), charcoal (n = 42), biofuels (n = 35), timber (n = 780), stems/cane (n = 115), fibres (n = 397), leaves (n = 58), seeds and fruits (n = 69), tannins and dyestuffs (n = 337), resins and gums (n = 101), lipids (n = 254), waxes (n = 26), scents (n = 165), rubber (n = 65), medicines (n = 1492), vertebrate poisons (n = 133), invertebrate poisons (n = 189), antifertility agents (n = 61), smoking materials and drugs (n = 199), symbolism, magic and inspiration (n = 185).

charcoal, biofuels), materials (wood, stems/cane, fibres, leaves, seeds/fruits, tannins/dyestuffs, gums/resins, lipids, waxes, scents, latex/rubber), medicines (human and veterinary), useful poisons (vertebrate poison, invertebrate poison) and social uses (antifertility agents, smoking materials/drugs, symbolic/magic/inspiration) (Fig 1) (see [11] for a full description of the categories). When more than one application of the same category was described for a given genus, we considered them as one single record. For example, if the fibre of a taxon is used to make mats, paper and cordage (three different applications), we simply recorded that the taxon is valuable as a source of fibre. The database includes a few extra categories that were either considered as "miscellaneous" in the Collection Standard or that showed very few records in *Mabberley's plant-book*. The use of leaves and seeds/fruits as materials are considered as "miscellaneous" in the Collection Standard, yet we took them up front as independent categories because we found many records that fit into these categories. The environmental

categories "erosion control", "revegetators", "soil improvers" and "agroforestry" described in the Collection Standard were merged into one single category (i.e. soil improvers) because they were very difficult to tease apart in many cases (e.g. many agroforestry plants also prevent soil erosion, and revegetators often improve soil quality). The same rationale was applied to the categories "shade/shelter" and "boundaries/barriers/supports" described in the Collection Standard, which were merged into one single category. We considered both realized (> 99% of the records) and mooted uses (as long as they were properly documented), and doubtful records were disregarded in any case. In addition to the binary matrix of plant uses, we assembled a control text matrix to link the former with the description of the uses in *Mabberley's plant-book*, and in those cases where more than one application of the same category was described for a given genus (see above) at least one of them was included in the text matrix (multiple entries were separated by "AND").

## Results

Our sampling procedure rendered a binary classification of 9478 plant-use records across 13489 genera of vascular plants and 28 categories of use. Of all the genera included in the dataset, 33.05% showed at least one benefit (hereafter "beneficial genera"), with a maximum number of records per genus of 17. Most beneficial genera (73.44%) provided just one or two types of services, the most common being "ornamental" (26%) followed by "medicines" (16%), "human food" (13%) and "timber" (8%), while the rest of benefits occurred at a frequency lower than 5% (Fig 1). The mean square contingency coefficient among the categories varied between -0.008 and 0.332, suggesting overall weak relationships among them.

## Discussion

Traditional knowledge on plant use is notably under-documented [16], which urges collective efforts to collate, validate and freely deliver ethnobotanical datasets to detect major sampling gaps and ensure evidence-informed policy making [17]. The retrieval of plant use information from *Mabberley's plant-book* and its arrangement as a readily available matrix for quantitative analyses is a major step towards achieving these goals and will help to advance scientific knowledge in a botanical discipline that is gaining momentum [18]. For example, a recent study drew on this database to show that phylogenetic diversity can efficiently capture plant services [11], supporting a promising macroevolutionary perspective on biodiversity conservation [19]. This is just one example of the potential of our database to expand scientific knowledge in the field and to achieve the ambitious biodiversity challenges that humanity must face in the coming decades [3].

The first edition of *Mabberley's plant-book* was published in 1987 and it has been updated and reissued every ten years since then, the necessary extensive changes in each edition making previous ones largely obsolete [15]. The 4th edition of the book, which was used to create the plant-use database presented here, was published in 2017, and a 5th edition could come sometime before 2030. Thus, the eventual publication of the 5th edition (in preparation) may represent a great opportunity for future growth of the database. It will come as no surprise that information on plant uses would be substantially increased in future editions of the book given the palpable demand for this type of information by scientists and policy makers [4]. Unfortunately, the preservation of ethnobotanical knowledge is severely endangered due to the strong cultural erosion that is linked to globalization and industrialization of human societies, which urges integrative policies to explicitly recognize the link between cultural and biological heritage [20]. The question is whether the ongoing international political commitments will

materialize fast enough as to overtake the alarming rates of plant extinction and culture loss we are witnessing [21].

The Intergovernmental Science-Policy Platform on Biodiversity and Ecosystem Services (IPBES) has recently recognized the need to assess the use of 'wild' species, including the identification of opportunities to promote sustainable practices [17]. The ultimate goal of this global conservation initiative is preserving the "option values" of biodiversity, this is, the myriad of present and yet-to-be discovered benefits that are associated with the continued existence of a high diversity of species in nature [22]. Our hope is that this database serves to help achieve this ambitious objective and thus preserve the beneficial potential of biodiversity for the future.

## Third party information

All icons in Fig 1 are adapted from the Noun Project (https://thenounproject.com) under a Creative Commons license CC BY 3.0; Yu luck, KR–In the Pollution solid Collection (radioactive waste) / Gregor Cresnar (warning sign) / Alice Design–In the Plant Tree Nature Leaf Eco Garden Natural Forest Collection (plant) (bioindicators and bioremediators); Ben Davis, RO–In the Smashicons Garden 2—Solid Collection (soil improvers); Wahyuntitle, ID (ornamental); Hamish–In the Environments & Nature Collection (hedges and shelters); By Icongeek26–In the Fruits and vegetables Collection (human food); Adrien Coquet, FR (food additives); H V P (vertebrate food); Tomi Triyana, ID (leaf) / Kiran Shastry, IN (silkworm) (invertebrate food); Vectors Market–In the Beach and Camping Glyph Icons Collection (fuelwood); Arthur Shlain, RU–In the Charcoal Collection (charcoal); By Icongeek26–In the Power Collection (biofuels); By Firza Alamsyah, ID–In the Autumn Collection (timber); By Symbolon, IT–In the Pixa Collection (stems/cane); iconixar–In the Sewing—Solid Collection (fibre); Hermine Blanquart, FR (leaves); kareemovic3000 (seeds and fruits); Marco Galtarossa, IT (tannins and dyestuffs); Bakunetsu Kaito–In the Construction Collection (resins and gums); Nikita Kozin, RU–In the Car Service Filled Collection (lipids); GeoNeo1, GB–In the Christmas Collection (waxes); DinosoftLab–In the Shopping and E-commerce Glyphs icons vol 1 Collection (scents); Xinh Studio–In the Simplie plants Collection (rubber tree) / Gan Khoon Lay–In the Farmer Farming Agriculture Plantation Industry Collection (rubber tapper) (rubber); Mavadee, TH–In the Hospital Collection (medicines); Diego Naive, BR (poison) / Peter van Driel, NL–In the An Icon I need tomorrow Collection (rat) (vertebrate poison); Diego Naive, BR (poison) / By SBTS, IN–In the Smartfarm BlackFill Collection (bug) (invertebrate poison); ProSymbols, US–In the Gynecology Glyph Icons Collection (antifertility agents); ProSymbols, US–In the Cells, Organs, Medical Cannabis Glyph Icons Collection (smoking materials and drugs); vanila–In the a Collection (yin yang) / ani rofiqah, ID–In the tree Collection (tree) (symbolism, magic and inspiration).

## Author Contributions

**Conceptualization:** Rafael Molina-Venegas, Miguel Ángel Rodríguez.

**Data curation:** Rafael Molina-Venegas, Manuel Pardo-de-Santayana, David J. Mabberley.

**Funding acquisition:** Rafael Molina-Venegas.

**Investigation:** Rafael Molina-Venegas.

**Methodology:** Rafael Molina-Venegas, Manuel Pardo-de-Santayana.

**Project administration:** Rafael Molina-Venegas.

**Supervision:** Rafael Molina-Venegas.

**Writing – original draft:** Rafael Molina-Venegas.

**Writing – review & editing:** Miguel Ángel Rodríguez, Manuel Pardo-de-Santayana, David J. Mabberley.

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
