## [Decision Letter · Decision Letter 0]

28 May 2021

A global database of plant services for humankind

PONE-D-21-14868

Dear Dr. Molina,

We’re pleased to inform you that your manuscript has been judged scientifically suitable for publication and will be formally accepted for publication once it meets all outstanding technical requirements.

Kind regards,

Narel Y. Paniagua-Zambrana, M.D.

Academic Editor

PLOS ONE

Additional Editor Comments (optional):

We want to mention to the authors that the decision that has been made in relation to this manuscript aims to highlight the contribution that the information and analysis carried out could generate to current ethnobotanical science. We believe that in the future this information could generate interesting analyzes, discussions and conclusions that could benefit the development of science and the making of decisions related to the conservation of species and the knowledge and use associated with them.

Reviewers' comments:

Reviewer's Responses to Questions

**Comments to the Author**

1. Is the manuscript technically sound, and do the data support the conclusions?

Reviewer #1: Yes

Reviewer #2: Yes

2. Has the statistical analysis been performed appropriately and rigorously? 

Reviewer #1: Yes

Reviewer #2: N/A

3. Have the authors made all data underlying the findings in their manuscript fully available?

Reviewer #1: Yes

Reviewer #2: Yes

4. Is the manuscript presented in an intelligible fashion and written in standard English?

Reviewer #1: Yes

Reviewer #2: Yes

5. Review Comments to the Author

Reviewer #1: Over the last months, this reviewer has already seen various editions of this manuscript, and the authors have always carefully incorporated all comments. This is an interesting manuscript indeed that will especially foment more discussion sin the discipline. From this perspective it does serve a wide audience.

It is time to getting this published!

Reviewer #2: This manuscript is an atypical contribution. It is about organising data in a better way for their use and does not generate new knowledge. I acknowledge that the work done is very important to us ethnobotanists, but I doubt PlosONE or any other such periodical is a suitable place for it unless there is a special section about databases.

You opted to submit your work to an Open Access journal, which probably means that you envisage spending a certain sum for its publication. Why not redirect this finances to creating a special webpage for your database? As an example, please have a look on clonal plant growth database: https://clopla.butbn.cas.cz/

You might also ask advice to national or international societies of ethnobotany how to make your database accessible and useful.

6. PLOS authors have the option to publish the peer review history of their article (what does this mean?). If published, this will include your full peer review and any attached files.

Reviewer #1: No

Reviewer #2: No

---

## [Editor Report · Acceptance letter]

7 Jun 2021

PONE-D-21-14868 

A global database of plant services for humankind 

Dear Dr. Molina-Venegas:

I'm pleased to inform you that your manuscript has been deemed suitable for publication in PLOS ONE. Congratulations! Your manuscript is now with our production department. 

Kind regards, 

on behalf of

Dr. Narel Y. Paniagua-Zambrana 

Academic Editor

PLOS ONE